# Biomimetic CO oxidation below −100 °C by a nitrate-containing metal-free microporous system

Konstantin Khivantsev [1,5] ✉, Nicholas R. Jaegers [1,5], Hristiyan A. Aleksandrov [2,5] ✉, Libor Kovarik[1], Miroslaw A. Derewinski [1,3], Yong Wang [1,4], Georgi N. Vayssilov [2] & Janos Szanyi [1,5] ✉

CO oxidation is of importance both for inorganic and living systems. Transition and precious metals supported on various materials can oxidize CO to $CO_2$. Among them, few systems, such as $Au/TiO_2$, can perform CO oxidation at temperatures as low as −70 °C. Living (an) aerobic organisms perform CO oxidation with nitrate using complex enzymes under ambient temperatures representing an essential pathway for life, which enables respiration in the absence of oxygen and leads to carbonate mineral formation. Herein, we report that CO can be oxidized to $CO_2$ by nitrate at −140 °C within an inorganic, nonmetallic zeolitic system. The transformation of $NO_x$ and CO species in zeolite as well as the origin of this unique activity is clarified using a joint spectroscopic and computational approach.

[1] Pacific Northwest National Laboratory, Richland, WA 99352, USA. [2] Faculty of Chemistry and Pharmacy, University of Sofia, Sofia 1126, Bulgaria. [3] Jerzy Haber Institute of Catalysis and Surface Chemistry, Polish Academy of Sciences, Krakow 30-239, Poland. [4] Voiland School of Chemical Engineering and Bioengineering Washington State University, Pullman, WA 99163, USA. [5] These authors contributed equally: Konstantin Khivantsev, Nicholas R. Jaegers, Hristiyan A. Aleksandrov, Janos Szanyi. ✉email: Konstantin.Khivantsev@pnnl.gov; Haa@chem.uni-sofia.bg; Janos.Szanyi@pnnl.gov

CO oxidation represents an important chemical transformation for both emissions control and living microorganisms[1–8]. To mitigate harmful CO emissions, inorganic materials such as transition- or noble metals supported on solid materials are capable of oxidizing CO at elevated temperatures[1,2]. Among such systems, Au nanoparticles supported on e.g., titania, discovered by Haruta, represents a material class active for CO oxidation at temperatures as low as −70 °C, the lowest presently known[1]. In microorganisms, enzymes evolved to oxidize CO to CO₂ under anaerobic and aerobic conditions using nitrate at room temperature, as shown by King and coworkers[3–5,8]. The energy produced is used to sustain life while the emitted CO₂ often leads to the formation of carbonate minerals[3–7], a key identifying characteristic that may help to understand the possible presence of anaerobic life on extraterrestrial bodies such as Mars[5,8]. Herein, a pathway for CO oxidation by nitrate in a completely inorganic, non-metallic crystalline solid (zeolite SSZ-13, CHA) is disclosed which is reactive at temperatures below −100 °C.

## Results

**NxOy chemistry in H-SSZ-13.** For a clear understanding of the behavior of CO in nitrate-containing aluminosilicate zeolite systems, the chemistry of NOx was first explored. In the 1980s, it was first discovered with Raman spectroscopy that NO⁺ species can form in Na-zeolites upon exposure to NO and O₂ or NO₂[9]. Later, Hadjiivanov and coworkers' pioneering studies using FTIR spectroscopy and isotopic methods on NxOy molecules[10–12] established, that NO⁺ can indeed form in H-ZSM-5 upon interaction with NO and O₂ mixture. NO₂ was shown to be disproportionate to NO⁺ and NO₃⁻ in the zeolites charged-balanced with monovalent Li, K, and Na as well[8]. Although, it was clear that NO⁺ was produced in zeolites from mixtures of NO and O₂ and mechanistic insight was proposed, we revisited its production and chemical properties using spectroscopy and density functional theory calculations.

NO₂ was first dosed onto a small pore H-SSZ-13 zeolite with a Si/Al ratio of ~12. SSZ-13 zeolite was selected due to its prevalence in the natural world[13]. SSZ-13 is a robust, hydrothermally stable framework used extensively to decrease pollution from vehicle operation[14–18] and it also contains only 1 equivalent framework T-site[15], reducing complexity in experimental interpretation and modeling relative to frameworks with a broad distribution of T-sites. Figure 1a depicts the in-situ FTIR data collected during sequential adsorption of NO₂ on H-SSZ-13 with Si/Al ~12 at 25 °C. Surprisingly, despite the fact that NO₂ interaction with alkali-substituted forms of zeolites has been

studied, no clear spectroscopic studies have been reported for NO₂ interaction with H-zeolites. During adsorption of nitrogen dioxide, two bands begin to grow simultaneously in the N–O stretching region, one with a maximum at ~2171 cm⁻¹ (NO⁺) and one with a maximum at ~1650 cm⁻¹ (NO₃⁻). This observation is consistent with the disproportionation reaction of NO₂ to NO and HNO₃ (equation (1)):

$$2NO_2(g) + H - Zeolite \rightarrow NO^+ - Zeolite + HNO_3(g) \quad (1)$$

Indeed, NO₂ is known to easily dimerize to N₂O₄ which, in highly polar solvents (ethyl acetate, for example) or in the highly polar zeolitic micropores, favors ionization to [NO⁺][NO₃⁻] fragments. By simple salt metathesis, these fragments can react with zeolite acid protons forming the species in a scheme depicted above.

The NO⁺ band at 2171 cm⁻¹ is asymmetric and it corresponds to two NO⁺ stretches at ~2195 and ~2171 cm⁻¹. Simultaneously, the two bands corresponding to acidic protons in the OH stretching region at 3612 and 3585 cm⁻¹ decreases (Fig. 1b), meaning that NO⁺ occupies the positions near Al atoms where those two distinct groups of protons were positioned. Concurrently, a new OH stretch appears at 3667 cm⁻¹, most likely corresponding to the -OH stretch of HNO₃ interacting with zeolite. Thus, NO₂ reacts with zeolite to form NO⁺ near the framework oxygens (in place of acidic protons) and HNO₃ fragments. Similar chemistry is observed for H-SSZ-13 with a different Si/Al ratio ~6 and a slightly different distribution with NO⁺ species formed (see difference spectra in Supplementary Fig. 1, HAADF-STEM images of SSZ-13 crystals are shown in Supplementary Fig. 2). Such a transition is consistent with the DFT calculations, that predict an exothermic transition from scheme 1 by −46 kJ/mol.

Conceptually, two types of NO⁺-X⁻ complexes may exist. (1) [N≡O]⁺ complexes with weakly coordinating anions like NO⁺BF₄⁻, in that NO⁺ behaves as a semi-free cation. Such complexes exhibit NO⁺ vibrational frequencies > 2300 cm⁻¹. For example, 2340 cm⁻¹ in NO[BF₄], 2326 cm⁻¹ in NO[AuF₆], and 2298 cm⁻¹ in concentrated sulfuric acid solutions for free [NO]⁺ [19]. (2) NO⁺-X⁻ complexes where a covalent bond is present between N and X, where X is often a halogen. In this case, O=N-X is bent with an O-N-X angle < 180° and NO vibrational frequencies are observed between 1950 and 1800 cm⁻¹ [20]. Even for non-halogens, the bent (120°–180°), covalent M-N=O moieties appear below 1900 cm⁻¹ in the IR spectra[21]. DFT calculations for free NO⁺ and NO⁺-X⁻ complexes are in good agreement with the experimental trend (Supplementary Table 1). However, for the NO⁺ in zeolite, NO⁺ frequencies are present in the intermediate region between

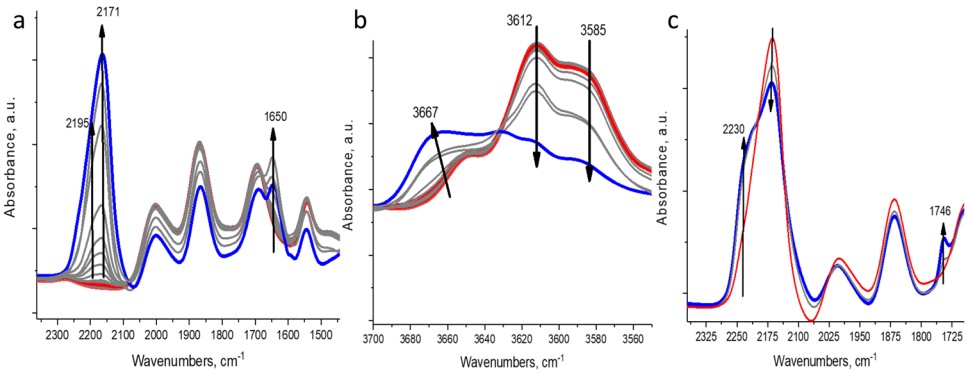

**Fig. 1 Monitoring NxOy species in H-SSZ-13 with FTIR. a** In-situ FTIR NO stretching region during sequential NO₂ adsorption (2 Torr) on H-SSZ-13 with Si/Al ~12. The **b** OH stretching region during the process is described in A. **c** Adsorption of excess NO₂ (2 Torr) on NO₂-saturated sample in A. IR cell vacuum was used as background (not zeolite itself) in **a**, **b**, and **c**.

2300 and 1950 cm$^{-1}$, closer to the former. This suggests that the NO$^+$ and O-Zeolite interaction has a significant covalent character and NO$^+$ is not a free-floating ion in the zeolite. This interpretation is supported by the observation of a constant band position and FWHM of NO$^+$ band upon heating the NO$^+$/SSZ-13 system from 77 to 298 K (Supplementary Fig. 3). Unlike the 2133 cm$^{-1}$ NO$^+$ band in ZSM-5[11], the NO$^+$ bands in SSZ-13 are located at higher wavenumbers (~2170 and 2200 cm$^{-1}$) and are characterized by higher thermal stability up to 200 °C under high vacuum (Supplementary Fig. 4) in contrast to NO$^+$ in H-ZSM-5 located at 2133 cm$^{-1}$ and which starts desorbing above room temperature.

Adsorption of excess NO$_2$ on NO$^+$/zeolite with Si/Al ~6 (Supplementary Fig. 5) and 12 (Fig. 1c) leads to the decrease of the NO$^+$ stretching band and the concomitant growth of an NO band at ~2230 cm$^{-1}$, that lies higher than the original NO$^+$ band by ~35 cm$^{-1}$ and ~58 cm$^{-1}$, respectively. Simultaneously, a new N-O stretch appears at ~1740 cm$^{-1}$ accompanied by another NO stretch at ~2080 cm$^{-1}$ which is more clearly seen in the difference spectra and spectra upon vacuuming (Supplementary Figs. 5 and 8). This indicates that the adsorption of excess NO$_2$ on NO$^+$ produces a NO$^+$-NO$_2$ complex. Such a complex has been suggested on the basis of synchrotron Rietveld refinement for NO$^+$ interacting with excess NO$_2$ in the super-cages of Ba-FAU zeolite[22]. The onset of this complex formation coincides with the appearance of the band at 1746 cm$^{-1}$ [23]. DFT calculations (Supplementary Fig. 10) predict the stretches of NO$^+$-NO$_2$ complex lie at ~2042 and 1722 cm$^{-1}$. Considering the parallel shift of calculated DFT NO stretches relative to experimental observations by 20–40 cm$^{-1}$, this agrees well with ~2080 and ~1746 cm$^{-1}$ bands for the NO$^+$-NO$_2$ complex. The nature of the 2230 cm$^{-1}$ band is discussed below.

Somewhat different chemistry is observed when reacting a mixture of NO and O$_2$ over zeolites. It was shown in the 1980s that NO and O$_2$ in microporous materials can easily produce NO$_2$[24]. The generated NO$_2$ can then either dimerize to form N$_2$O$_4$ or react with NO to form N$_2$O$_3$. Due to the presence of excess NO in the system, the primary reaction is N$_2$O$_3$ formation. The thus formed N$_2$O$_3$ can disproportionate to the ion pair NO$^+$NO$_2^-$ that, in turn, can react with a zeolitic proton. Indeed, when we mix excess of NO with sub-stoichiometric quantities on oxygen, we see the growth of IR bands characteristic of both NO$^+$ and N$_2$O$_3$ features (Supplementary Fig. 6). Interestingly, the N$_2$O$_3$ feature quickly grows but then begins to slowly decompose concomitant with the increase of the NO$^+$ feature (Supplementary Fig. 6). N$_2$O$_3$ (1570 and 1970 cm$^{-1}$) consumes Brønsted acid protons according to Equation (2):

$$N_2O_3 \leftrightarrow NO^+ + NO_2^-$$

$$NO^+NO_2^- + H - Zeolite \rightarrow NO^+ - Zeolite + HNO_2 \quad (2)$$

HNO$_2$ (NO$^+$OH$^-$) then quickly reacts with H$^+$ to form NO$^+$ and H$_2$O. In the sub-stoichiometric O$_2$ case (Scheme 2), NO combines to form predominantly NO$^+$ and H$_2$O. In the case of NO$_2$ (Scheme 1), NO$^+$ and NO$_3^-$ form. NO$^+$ is stable up to 150 °C under a high vacuum. Adsorption of NO$_2$ at room temperature leads to the formation of the NO$^+$-NO$_2$ complex and shifts the NO$^+$ band to 2230 cm$^{-1}$. Evacuation restores the original NO$^+$ and removes the bands of [NO$^+$-NO$_2$] complex (Supplementary Figs. 7 and 8). Furthermore, at 100 K, adsorption of NO selectively produces the NO$^+$-NO complex at 2013 cm$^{-1}$.

**CO chemistry in H-SSZ-13**. Adsorption of CO at low temperature (100 K) shows similar complexation, forming the NO$^+$-CO complex and blue shifting the NO$^+$ band to ~2220 cm$^{-1}$ (Fig. 2a). Furthermore, at this temperature (100 K) adsorption of NO

NO unambiguously and selectively produces the NO$^+$-NO complex. (Figure 2b). In contrast, upon adsorption of NO on the NO$^+$/H/SSZ-13 sample at 100 K the IR band of NO$^+$ redshifts to 2013 cm$^{-1}$ evidencing the selective production of the NO$^+$-NO complex.

This observation provides new insight into how the adsorption of an adsorbate (NO, CO, and NO$_2$) changes the properties of the cation with which it interacts. Normally, such cations are metal cations that present no corresponding active stretches, hiding information regarding the changes incurred during adsorption. In the system presented herein, NO$^+$ provides such insight due to the active N–O vibration.

Our DFT calculations show that the interaction between NO$^+$ and CO or NO$_2$ is relatively weak with a binding energy of the adsorbed molecule below −20 kJ/mol (Supplementary Table 1). We conclude that since the concentration of CO or NO$_2$ is high in the zeolite, these gas-phase molecules slightly shift NO$^+$ further from its equilibrium position, slightly weakening the interaction between NO$^+$ and zeolite. This leads to shortening in the N-O distance and shift of N-O vibrational frequency to higher wavenumbers as evidenced by the shift of the NO band to ~2230 cm$^{-1}$ (NO$_2$) and 2220 cm$^{-1}$ (CO). This perturbation provides a unique insight into the interaction of extra-framework cations with adsorbates not routinely available even from the most sophisticated synchrotron XRD and Rietveld refinement methods. In the case of NO, however, the shift is to the significantly lower frequencies. This finding can be rationalized by our DFT results, since in this case a stable NO$^+$-NO/Zeolite complex is formed (Supplementary Fig. 10) in the zeolite with a binding energy of NO to NO$^+$/Zeolite of −52 kJ/mol (Supplementary Table 1). This structure has two frequencies at 2009 and 1911 cm$^{-1}$. These can rationalize the experimental bands at 2013 and 1870 cm$^{-1}$. In addition, calculated ONNO species in the gas phase have frequencies at 1879 and 1727 cm$^{-1}$, rationalizing the experimental bands at 1870 and 1685 cm$^{-1}$.

**CO oxidation to CO$_2$ at low temperature**. In-situ heating of the NO–CO complex produced from N$_2$O$_3$ reaction with H-SSZ-13, leads to no CO$_2$ formation (Supplementary Fig. 9). However, when the CO/nitrate system (from NO$_2$ disproportionation, Scheme 1) temperature is raised from −170 to −140 °C, an immediate formation of CO$_2$ is observed inside zeolite micropores with a sharp characteristic 2345 cm$^{-1}$ signature[25,26]. Simultaneously, the nitrate band decreased due to its consumption, as shown in Fig. 3. This result indicates that CO is oxidized by nitrate to form CO$_2$ at the low temperature of −140 °C (Fig. 3).

This biomimetic chemistry by a completely inorganic system occurs at temperatures previously unattainable for such a conversion. Moreover, when CO with NO$_3^-$ is reacted at room temperature (in the same system), no reaction is observed. To explain this, we note that catalysis occurs when the reacting molecules are adsorbed/ chemisorbed to an active center. At room temperature, CO is not adsorbed by the Brønsted acid protons of -Si-OH-Al groups, as evidenced by the lack of CO stretches other than gas-phase CO. However, CO interacts with Brønsted acid protons of the zeolite at lower temperatures forming -H$^+$-CO complex[24–26]. The IR CO stretch in this complex is 2175 cm$^{-1}$ [24–26] and CO is polarized as C$^{(\delta+)}$-O$^{(\delta-)}$ since no back donation is present. Only electrostatic interactions and the formation of a sigma bond takes place with charge transfer from C to H$^+$, making CO susceptible to nucleophilic attack by NO$_3^-$ to form CO$_2$ and reduced N species. DFT calculations further support this proposed route. We considered four mechanisms for CO oxidation by HNO$_3$ in H-CHA. They are presented schematically in Fig. 4 in addition to the corresponding energetic diagrams.

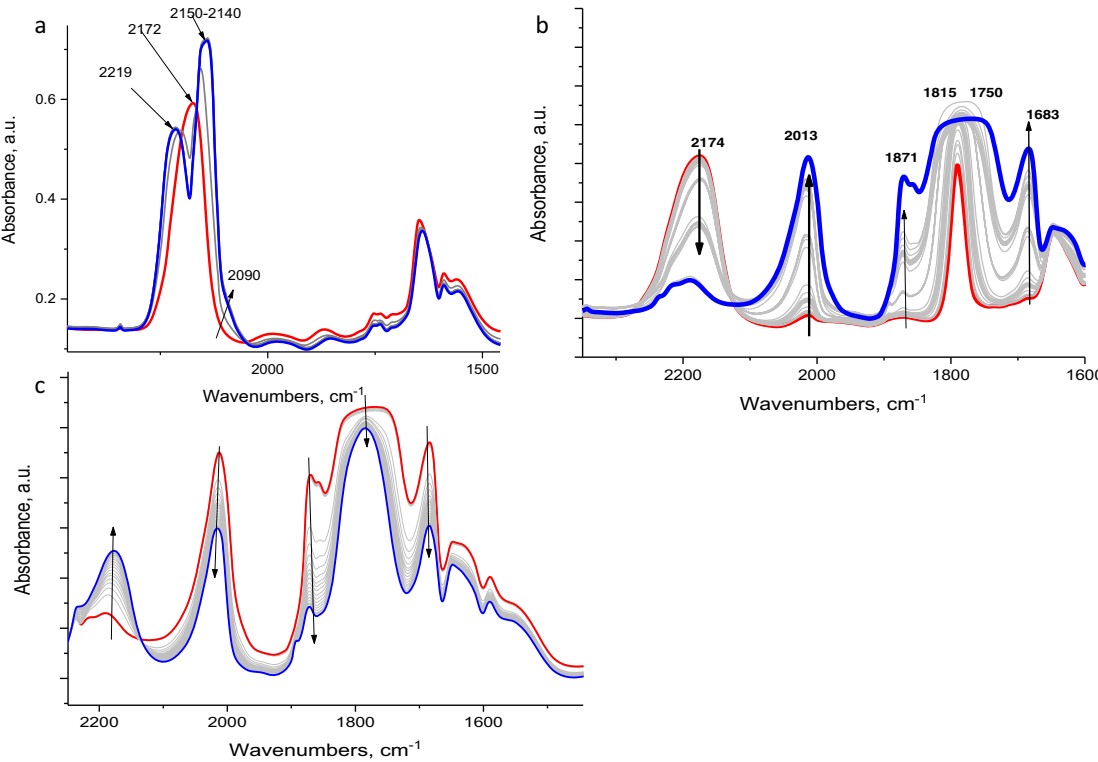

**Fig. 2 Monitoring NO⁺ interaction with adsorbates. a** In-situ FTIR during sequential CO adsorption (2 Torr) at 100 K on H-SSZ-13 (Si/Al ~12) which was previously reacted with $NO_2$ at RT to form $NO^+$ and $NO_3^-$. Excess of $NO_2$ was removed by vacuuming at $10^{-6}$ Torr at RT. **b** In-situ FTIR during sequential NO adsorption (2 Torr) at 100 K on H-SSZ-13 (Si/Al ~12) that was previously reacted with $NO_2$ at RT Excess of $NO_2$ was removed by vacuuming at $10^{-6}$ Torr at RT. **c** In-situ FTIR during $10^{-7}$ Torr vacuum applied after B at 100 K. NO leaves Zeo-$NO^+$-NO complex under vacuum, restoring $NO^+$. The zeolite pellet itself was used as an IR background.

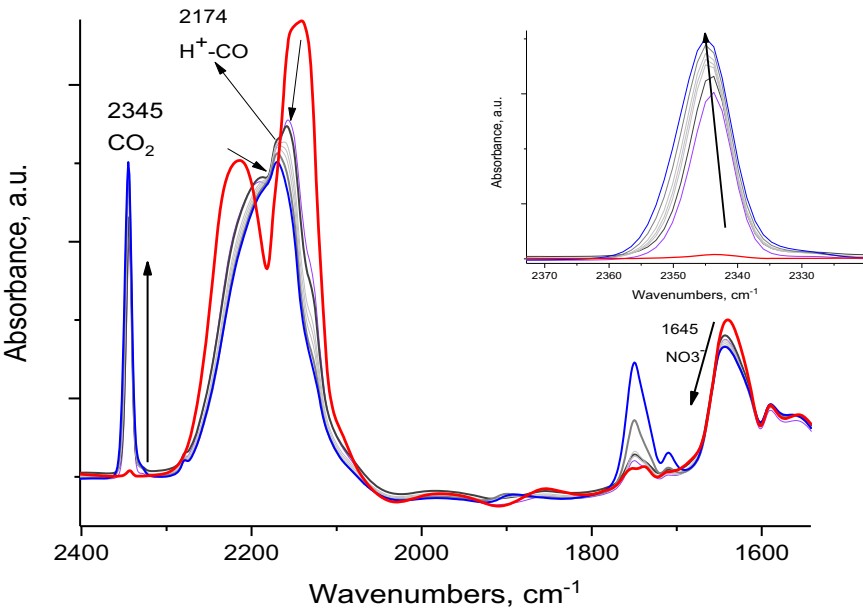

**Fig. 3 Monitoring CO oxidation with FTIR.** In-situ *FTIR* during an increase of temperature from 100 K (red line, described in Fig. 2a, H-SSZ-13 with Si/Al ~12) to 135 K. At this temperature $CO_2$ starts to evolve and the temperature is held until $CO_2$ reaches the maximum level (~2 min). The inset shows the magnified $CO_2$ region. Note that we collected spectra at temperatures intermediate between 100 and 135 K, and no $CO_2$ formation was detected until 135 K. Zeolite pellet itself was used as IR background.

## Discussion

All mechanisms start with the adsorption of the reactants, CO and $HNO_3$, via hydrogen bonds to the bridging acidic OH group and basic zeolite oxygen center, respectively. In this initial state (IS) structure (Zeo1ite/CO/$HNO_3$) the C–O vibrational frequency was calculated to be 2175 cm⁻¹, in line with the corresponding experimental band. Mechanisms A and B are one-step mechanisms. In the former, the CO molecule is oxidized by the $HNO_3$ via

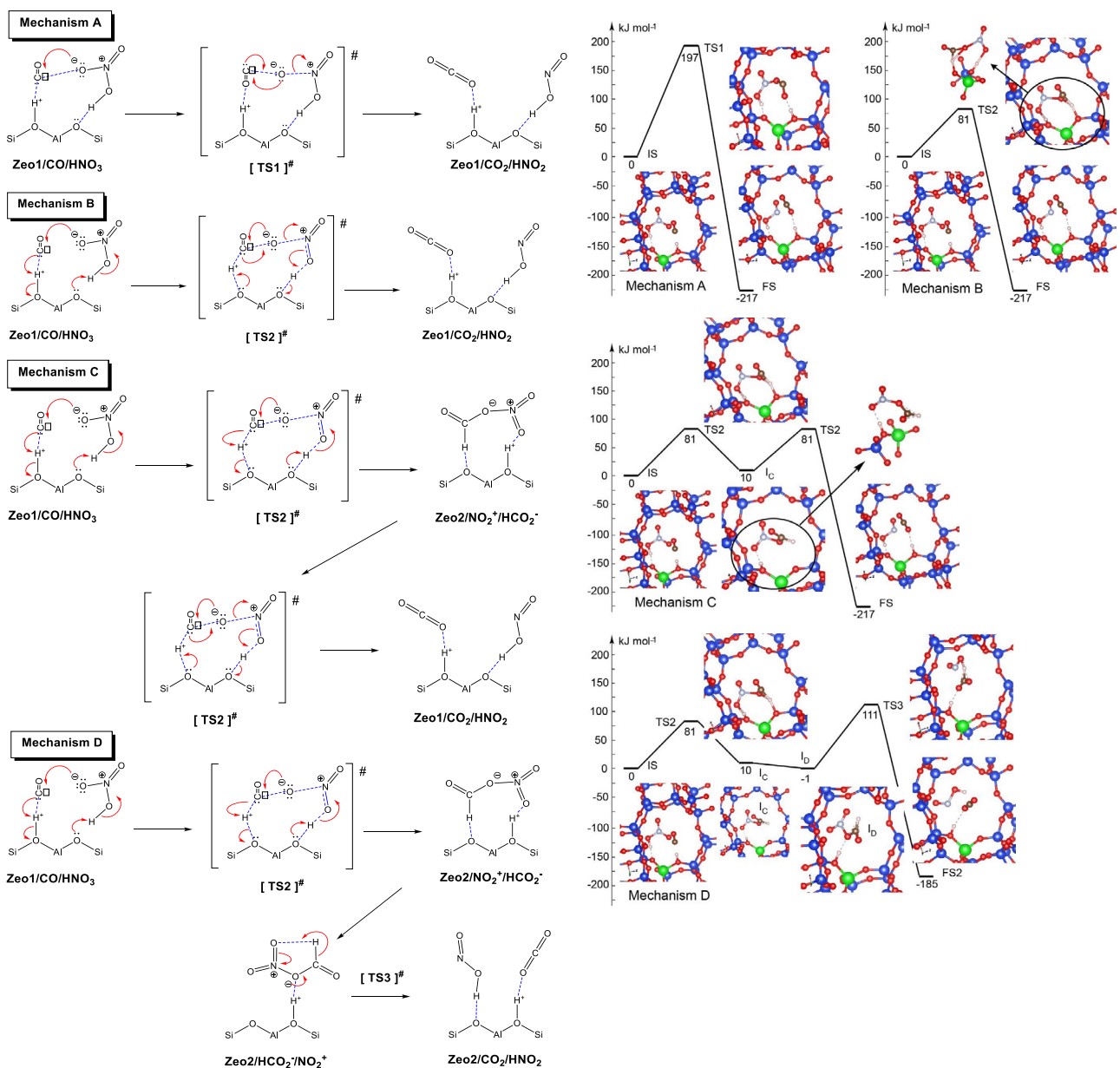

**Fig. 4 Mechanisms A, B, C, and D of CO oxidation by HNO₃ in H-CHA.** Zeolite structures Zeo1 and Zeo2 differ by the different locations of the zeolite proton at basic O centers from the AlO₄ unit. The first species given after Zeo1 and Zeo2 are the one which interacts with the zeolite proton. Corresponding energy diagrams of the investigated mechanisms (Mechanisms A, B, C, and D) for CO oxidation by HNO₃ in H-CHA. The corresponding structures are also shown. Shorter names are used for the structures in the energy diagrams with respect to the schemes above as: IS – Zeo1/CO/HNO₃; FS – Zeo1/CO₂/HNO₂; I$_C$ – Zeo2/NO₂⁺/HCO₂⁻; I$_D$ – Zeo2/HCO₂⁻/NO₂⁺, FS2 – Zeo2/CO₂/HNO₂. Color coding: Si – blue, O – red, Al – green, N – light blue, C – brown, H – white.

transition state structure TS1 with no direct participation of the zeolite. In mechanism B, the CO molecule is oxidized to CO₂ via transition state structure TS2, where the zeolite participates actively in the process via movement of the proton from the bridging OH group towards the C atom of the CO molecule and via the basic zeolite oxygen, which attracts the nitric acid's proton. TS2 structure seems to correspond to a bifurcation point, since it may be decomposed in two ways (i) directly to CO₂ and HNO₂ via the electron transfer shown by the arrows in the mechanism B; and (ii) to the formation of a complex HCOO⁻NO₂⁺ (I$_C$) as the zeolite proton interacts with an O center from NO₂⁺, as shown in mechanism C. From the latter intermediate, the final products (CO₂ and HNO₂) can be formed via the same TS2 structure (Mechanism C). Alternatively, if the intermediate is

coordinated to the zeolite (see Zeo2/HCO₂⁻/NO₂⁺, I$_D$) CO₂ and HNO₂ can be formed via TS3, that includes an H⁻ transfer from HCOO⁻ to NO₂⁺ (Mechanism D). In all mechanisms considered the CO oxidation was calculated to be exothermic by 187 to 217 kJ/mol, depending on the coordination of the products (Fig. 4), thus we investigated only kinetically relevant aspects in detail. Energy diagrams show that the most plausible are Mechanisms B and C with a barrier of the rate-limiting steps (which are the same) of 81 kJ/mol. In addition, we calculated the Gibbs free energy barriers of this rate-limiting step of both mechanisms at 140 K to be 58 kJ/mol. Since these steps are the first ones, observation of an intermediate is not expected in line with our experimental results. Furthermore, the product of the reaction (HNO₂) formation is inferred additionally on the basis of

FTIR data (Supplementary Fig. 12 depicting the HON bending region with the band ~1260 cm$^{-1}$)[27].

Both mechanisms involve the direct participation of the zeolite via H$^+$ transfers and as well as the formation of an initial complex of CO to the zeolite H$^+$ is crucial for the oxidation process. The calculated energy barriers for the rate-limiting steps of mechanisms A and D are notably higher (Fig. 4).

We also considered CO oxidation on NO$_2^+$/Zeolite (Supplementary Fig. 11), in which NO$_2^+$ are the charge-compensating species of the negatively charged zeolite framework, forming a nitrate species employing the O center from the zeolite. The calculated IR frequencies of such NO$_2^+$/Zeolite species are 2046 and 1358 cm$^{-1}$ and move to 2055 and 1353 cm$^{-1}$ after CO adsorption. However, the predicted barrier for the reaction: NO$_2^+$/Zeolite + CO → NO$^+$/Zeolite + CO$_2$ is prohibitively high, 143 kJ/mol, discarding the role of NO$_2^+$ species as a CO oxidizing agent.

In summary, it is shown that CO can be oxidized to CO$_2$ by nitrate in zeolite micropores at temperatures as low as −140 °C. This reaction was previously known to be catalyzed by complex enzyme molecules in living (an)aerobic organisms. However, no metallic systems have been known to perform this biologically important reaction under mild conditions. Notably, a fully inorganic system with no (noble) metals performs such a reaction at −140 °C. Interaction of CO with Brønsted acid protons in confined nanopore produces an -H$^+$-CO complex making the carbon atoms susceptible for nucleophilic attack by a nitrate, opening a hitherto unknown pathway for CO conversion chemistry in inorganic systems at low temperatures.

## Methods

**Synthesis**. H-SSZ-13 was synthesized using well-established synthesis methods[14,16,18].

First, Na-SSZ-13 was hydrothermally synthesized. The composition of the gel is as follows: 10SDA:10-NaOH:xAl$_2$O$_3$:100SiO$_2$:2200H$_2$O, where x varies to allow the preparation of samples with different Si/Al ratios. The gel is prepared first by dissolving NaOH (99.999%, Aldrich) in water and adding the SDA (TMAda-OH, Sachem ZeoGen 2825). Following this, Al(OH)$_3$ (54% Al$_2$O$_3$, Aldrich) and fumed silica (7 micrometers average particle size, Aldrich) were added sequentially under continuous stirring. The gel was subsequently loaded into a 125 ml Teflon-lined stainless steel autoclave. Hydrothermal synthesis was performed at 160 °C for 96 hours under stirring. After synthesis, Na/SSZ-13 was separated from the mother liquid via centrifugation and washed with deionized water 3 times. Finally, the zeolite powders were dried at 120 °C under dry nitrogen, and calcined in dry airflow for 5 h at 550 °C.

More specifically, a Na-form of SSZ-13 with Si/Al ratio~6 and ~12 were prepared, and then exchanged three times with 2 M ammonium nitrate solution at 80 °C. The powder was purified by consecutive washing and centrifugation cycles with de-ionized water. The wet powder was dried at 100 °C, and then calcined at 550 °C for 5 h in the flow of dry air.

**Characterization**. The in situ static transmission IR experiments were conducted in a home-built cell housed in the sample compartment of a Bruker Vertex 80 spectrometer, equipped with an MCT detector and operated at 4 cm$^{-1}$ resolution. The powder sample was pressed onto a tungsten mesh which, in turn, was mounted onto a copper heating assembly attached to ceramic feedthrough. The sample could be resistively heated or cooled with liquid nitrogen, and the sample temperature was monitored by a thermocouple spot welded onto the top center of the W grid. The cold finger on the glass bulb containing CO was cooled with liquid nitrogen to eliminate any contamination originating from metal carbonyls, while NO was cleaned with multiple freeze–pump–thaw cycles. Prior to spectrum collection, a background with the activated (annealed, reduced, or oxidized) sample in the IR beam was collected. Each spectrum reported is obtained by averaging 64 scans.

HAADF-STEM was performed with an FEI Titan 80–300 microscope operated at 300 kV. The instrument is equipped with a CEOS GmbH double-hexapole aberration corrector for the probe-forming lens, which allows for imaging with 0.1 nm resolution in scanning transmission electron microscopy mode (STEM). The images were acquired with a high angle annular dark-field (HAADF) detector with an inner collection angle set to 52 mrad.

**Computational details and models**. Periodic DFT calculations were performed using the Perdew-Burke-Ernzerhof (PBE)[28], exchange-correlation functional with

the additional empirical dispersion correction D2 as proposed by Grimme (denoted as PBE + D2)[29] as implemented in Vienna ab initio simulation package (VASP)[30,31]. Test calculations were also done by using Heyd–Scuseria–Ernzerhof (HSE06) functional[32,33] and Strongly constrained and appropriately normed semilocal density functional (SCAN)[34] (see below). We also employed PAW pseudopotentials[35,36] and the valence wave functions were expanded on a plane-wave basis with a cutoff energy of 415 eV. The Brillouin zone was sampled using only the Γ point[37].

We used a monoclinic unit cell of the CHA framework, which consists of 36 T atoms. It was optimized for the pure silicate structure with dimensions: a = b = 13.675 Å, c = 14.767 Å; α = β = 90°, and γ = 120°[38]. One Si center in the unit cell located in one six-member ring was replaced with Al, as the negative charge around this Al was compensated by an H$^+$ cation. During the geometry optimization, atoms were allowed to relax until the force on each atom became less than $5 \times 10^{-2}$ eV/Å.

The vibrational frequencies were calculated using a normal mode analysis where the elements of the Hessian were approximated as finite differences of gradients, displacing each atomic center by $1.5 \times 10^{-2}$ Å either way along each Cartesian direction.

The reported binding energies (BE) of the various adsorbates (CO, CO$_2$, NO, HNO$_3$, and HNO$_2$) were calculated as BE = − E$_{ad}$ – E$_{sub}$ + E$_{ad/sub}$, where E$_{ad}$ is the total energy of the adsorbate in the gas phase (ground state), E$_{sub}$ is the total energy of the pristine zeolite system, where the framework negative charges are compensated by some of the modeled cationic species (H$^+$, NO$^+$, or NO$_2^+$) considered, and E$_{ad/sub}$ is the total energy of the zeolite, together with the adsorbate in the optimized geometry. With the above definition, negative values of BE imply a favorable interaction.

When Gibbs free energies were obtained the enthalpy values were calculated from the total energy values (E$_{el}$) corrected for the internal vibrational energy (E$_v$)[36] and zero-point vibrations (ZPE) derived from frequency calculations of the optimized structures: H = E$_{el}$ + E$_v$ + ZPE.

The entropy values of the initial and transition states (TS) include only the vibrational degrees of freedom (S$_v$), since the adsorbates are bound to the zeolite and the rotational and translational degrees of freedom are converted into vibrations[39–41]. The expressions of all enthalpy and entropy contributions can be found elsewhere[42].

**Verification of the employed PBE + D2 functional**. In order to investigate the suitability of the employed GGA functional (PBE + D2) we also did test calculations with one hybrid (HSE06) and one meta-GGA (SCAN) functional (Supplementary Tables 2 and 3). The initial structures were taken from geometry optimized with functional PBE + D2 and were reoptimized with HSE06 or SCAN functions.

The calculated total binding energy of CO and HNO$_3$ molecules in the ZEO1/CO/HNO$_3$ structure was similar with the three functionals: −116 kJ/mol, −117 kJ/mol, and −99 kJ/mol, respectively at PBE + D2, HSE06, and SCAN. We also optimized the Zeo/NO$^+$-NO and Zeo/NO$^+$-NO$_2$ complexes with the three functionals. The BE of NO in Zeo/NO$^+$-NO calculated at PBE + D2 level, −21 kJ/mol, was similar to the value at HSE06 level, −19 kJ/mol, while stronger interaction (−37 kJ/mol) was predicted by SCAN. For Zeo/NO$^+$-NO complex the BE values were identical, −52 kJ/mol, for PBE + D2 and SCAN functionals, while much weaker interaction, −21 kJ/mol, in the complex was calculated at HSE06 level. So, the calculated energy differences between the meta-GGA and the hybrid functionals are of the same order as the difference between each of them and the GGA functional with dispersion correction.

We also compared computational results for reaction energy of the CO oxidation by HNO$_3$ in the gas phase to CO$_2$ and HNO$_2$ with experimental value derived from the standard experimental enthalpies, −225 kJ/mol. The calculated values obtained with PBE + D2, HSE06, and SCAN functionals are −226 kJ/mol, −227 kJ/mol, and −236 kJ/mol, respectively, i.e., all three functional give very close values to the experiment. This comparison also suggests that the computational approach, used in the present work is relevant for the studied process.

A comparison of the calculated vibrational frequencies with the three functionals showed that the PBE + D2 results were notably closest to our experimental bands (the difference of the individual bands is 4–40 cm$^{-1}$), while HSE06 and SCAN frequencies are higher by 94–180 cm$^{-1}$ and 254–347 cm$^{-1}$, respectively. As a benchmark, we also calculated C–O vibrational frequency of the CO molecule in gas phase. Again the PBE + D2 value, 2130 cm$^{-1}$, was closest to the experimental band, 2143 cm$^{-1}$, while HSE06 and SCAN values were higher by 87 and 151 cm$^{-1}$ than the experimental value.

In order to check whether the proposed most plausible reaction pathway for CO oxidation by HNO$_3$ in the zeolite is functional dependent, we calculated all the barriers at the HSE06 level. Despite that the overall reaction energies calculated with PBE + D2 and HSE06 functionals are very close, within 1−9 kJ/mol, all HSE06 barriers are systematically shifted to higher values by 50−80 kJ/mol in comparison to values calculated with PBE + D2 functional. A similar trend has been reported in the recent theoretical study of Goncalves et al.[41]. However, at both levels of theory the order of barrier heights was the same: TS2 > TS3 > TS1, hence both sets of data lead to same conclusions for the plausible reaction mechanism.

## Data and materials availability

All data is available in the main text or the Supplementary Information.

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

## Acknowledgements

GNV is grateful for the support by the National scientific program "Information and Communication Technologies for a Single Digital Market in Science, Education and Security (ICTinSES)", financed by the Bulgarian Ministry of Education and Science. HAA is grateful for the support by the European Regional Development Fund and the Operational Program "Science and Education for Smart Growth" under contract UNITe № BG05M2OP001-1.001-0004 -C01 (2018–2023).

The research at PNNL was supported by the U.S. Department of energy, office of basic energy sciences, division of chemical sciences, biosciences, and geosciences catalysis program (DEAC05-RL01830, FW-47319). Experiments were conducted in the environmental molecular sciences laboratory (EMSL), a national scientific user facility sponsored by the Department of energy's office of biological and environmental research at Pacific Northwest National Laboratory (PNNL). PNNL is a multi-program national laboratory operated for the DOE by Battelle Memorial Institute under Contract DE-AC06-76RL01830.

## Author contributions

K.K. and J.S. conceived the idea and wrote the paper. K.K., N.R.J., M.A.D., and J.S. performed all the synthesis and characterization experiments. K.K., N.R.J., M.A.D., and JSz analyzed all the data. L.K. performed imaging. H.A.A. and G.N.V. performed all the DFT calculations and cowrote the DFT sections in the paper. Y.W. obtained funding. K.K., N.R.J., M.A.D., J.S., L.K., H.A.A., Y.W., and G.N.V. discussed all the data.

## Competing interests

The authors declare no competing interests.
