## [Peer Review File · Nature Communications]

Title: Biomimetic CO oxidation below $-100\text{ }^{\circ}\text{C}$ by a nitrate-containing metal-free microporous systemREVIEWER COMMENTS

Reviewer #1 (Remarks to the Author):

In this manuscript, Khivantsev et al. reports an unexpected oxidation reaction of CO by nitrate at a low temperature of 135 K in the H-SSZ-13 zeolite. In-site FT-IR was extensively employed to monitor the binding modes of NO_x in the presence and absence of CO in the H-SSZ-13. The spectroscopic data suggests that CO could be oxidized into CO₂ by NO₂ in the H-SSZ-13 at 135 K, while similar oxidation cannot be observed at room temperature. Due to its fundamental and practical importance, room-temperature CO oxidation is of current interest. However, an oxidation reaction that can only occur at low temperature (around 135 K) has limited potential for industrial use, as cooling a reaction system could be even more energy-consuming than heating the reaction system. Because of this significance problem together with a number of existing technical issues (see details below), I cannot recommend acceptance of this manuscript in its current form for publication.

1. Regarding the structure of this manuscript, the authors extensively discussed the formation and complexation processes of NO⁺ species in H-SSZ-13 in the first half of the manuscript, while the second half elaborated the oxidation of CO by NO₂. However, the proposed mechanism of CO oxidation has little to do with NO⁺ (Figure 4). Please the authors illustrate the role of NO⁺ in the NO₂ oxidation of CO.
2. Figure 1 exhibits the FT-IR spectra of H-SSZ-13 (with Si/Al ~ 12) after NO₂ adsorption. The peaks at ~1870 and 2100 cm⁻¹ should be assigned. Also, I find these peaks become less distinguishable in Si/Al ~ 6 sample (Figure S1). Such difference should be rationalized.
3. In Figure 1, the experimental parameters corresponding to individual line (including red, blue and grey lines) should be detailed. Similar figure legend issue exists almost in every FT-IR spectrum (Figure 2, 3, S1, S4, S5, S6, S7, S8 and S9), which makes some figures (e.g., Figure S6 and S8) almost unreadable.
4. What is the Si/Al ratio of H-SSZ-13 reported in Figure 3? There are two types of H-SSZ-13 with different Si/Al ratios (i.e., 12 and 6) investigated in this study, and the CO oxidation performance of these two types of H-SSZ-13 should be reported and compared.
5. In the caption of Figure 2, the adsorption temperature was quoted as 100 K, while that mentioned in the main text (Lines 133-136) was 77 K. Such discrepancy should be addressed.
6. Lines 136-137: It reads "... adsorption of NO unambiguously and selectively produces the NO⁺-NO complex. Figure 3A)". However, Figure 3 has nothing to do with NO adsorption.

Reviewer #2 (Remarks to the Author):

This paper reports that NO₂ adsorption on zeolite disproportionate to form a NO⁺/NO₃⁻ complex, and that NO₃⁻ acted as an oxidant for CO to produce CO₂ at -140 degree C. The reaction mechanism was discussed by in-situ FTIR and DFT calculation. The CO oxidation proceeded in the absence of noble metal nanoparticles or transition metal oxides at very low temperature, although the reaction stopped at high temperature due to desorption of CO from zeolite surface at high temperature. The phenomenon is very interesting but the discussion on FTIR results is still unclear. Therefore, I think that revision is required for the following points.

1) In Figure 2, several wavenumbers are shown but assignment was not completed for all the peaks mentioned in the figure. Figure captions should be clearly written to indicate which color corresponds to which state (e.g., NO⁺/NO₃⁻ complex, after CO adsorption, etc.). In Figure 2A, the peak at 2172 cm⁻¹ was assigned to NO⁺ in NO⁺/NO₃⁻ and was shifted to 2219 cm⁻¹ after CO adsorption. Although the authors did not mention the peak corresponding to NO₃⁻ (1650 cm⁻¹ from Fig. 3) in this figure, the NO₃⁻ peak seems to not change. Why only NO⁺ species was affected by CO adsorption while DFT calculation suggests that NO₃⁻ is located near the CO adsorbed on the zeolite. The peak at 2140-2150 cm⁻¹ should be mentioned.

2) In Figure 2B, the authors discussed the NO adsorption onto NO₂-preadsorbed zeolite and the formation of NO⁺/NO complex. I could not follow how the NO⁺/NO complex formation relates to the CO oxidation using nitrates adsorbed on zeolites.

In p. 5, line 6 from the bottom, Figure 3A should be Figure 2B?

3) In Figure 3, CO is oxidized to CO₂, NO⁺ peak was shifted to 2172 cm⁻¹ again, indicating CO was consumed, and the peak corresponding to CO₂ was newly observed. At the same time, the NO₃⁻ peak (1650 cm⁻¹) decreased, indicating NO₃⁻ play a crucial role for CO oxidation. According to proposed mechanism shown in pp. 9-10, HNO₂ is supposed to be formed after the CO oxidation. Did they observe the formation of HNO₂ by FTIR? Again, the peak at ca. 1750 cm⁻¹ was increased by not explained the reason.

4) In p. 8, the authors mentioned that CO is adsorbed at the Bronsted acid sites. Is there any possibility that CO is adsorbed at the Lewis acid sites on the zeolite?

5) In Zeo1/CO/HNO₃ in Mechanism C, H is missing.

Reviewer #3 (Remarks to the Author):

This manuscript reports the discovery of low-T CO oxidation in NO₂-activated SSZ-13 zeolites using FTIR and DFT studies, along with proposed mechanisms and energetics of CO oxidation. The manuscript (with the exception of few grammatical errors and incomplete sentences) clearly lays out the details of the formation of the active NO⁺ species on the zeolite and CO adsorption characteristics. The work is new and interesting.

My primary concern lies with the DFT studies since they play a large role in interpretation of experimental results. While it is encouraging that DFT results agree with experiment in general, the authors must note that PBE+dispersion may not be the best choice of theory for SSZ-13 (See:

<https://doi.org/10.1002/cctc.201900791>). This is especially important for binding energies, where it is necessary to verify whether CO indeed binds weakly to the zeolite-bound NO+. This will also help further verify the proposed mechanism(s) by examining whether the energetics of the four pathways are sensitive to the choice of level of theory.

However, DFT errors vary with the properties as well (vibrational frequencies, binding energies), making it difficult to choose the most physically appropriate level of theory. I therefore recommend the authors perform at least a small subset of all calculations using a functional more suited for these systems (e.g. hybrid functionals, SCAN) to verify that their interpretation of experimental observations are still valid.

Minor concerns:

1. In Figure 1, how do the authors interpret the new OH stretch at 3667cm⁻¹?
2. Line 53 – Sentence is incomplete
3. Scheme 1 is not charge-balanced (H⁺-zeolite on LHS?)

Response to reviewers' comments

Manuscript Number: NCOMMS-20-30796.

Title: Biomimetic CO oxidation below $-100\text{ }^{\circ}\text{C}$ by a nitrate-containing metal-free microporous system

Author(s): Konstantin Khivantsev, Nicholas R. Jaegers, Hristiyan A. Aleksandrov, Libor Kovarik, Mirosław A. Derewinski, Yong Wang, Georgi N. Vayssilov and Janos Szanyi

Reviewer #1 (Remarks to the Author):

In this manuscript, Khivantsev et al. reports an unexpected oxidation reaction of CO by nitrate at a low temperature of 135 K in the H-SSZ-13 zeolite. In-site FT-IR was extensively employed to monitor the binding modes of NO_x in the presence and absence of CO in the H-SSZ-13. The spectroscopic data suggests that CO could be oxidized into CO₂ by NO₂ in the H-SSZ-13 at 135 K, while similar oxidation cannot be observed at room temperature. Due to its fundamental and practical importance, room-temperature CO oxidation is of current interest. However, an oxidation reaction that can only occur at low temperature (around 135 K) has limited potential for industrial use, as cooling a reaction system could be even more energy-consuming than heating the reaction system. Because of this significance problem together with a number of existing technical issues (see details below), I cannot recommend acceptance of this manuscript in its current form for publication.

Response: We thank the Reviewer for taking the time to evaluate our manuscript and we appreciate both positive and critical comments. We have modified the manuscript in accordance with the Reviewer's comments and suggestions further strengthening the manuscript.

The Reviewer suggests that because our study is of no practical importance for the field of industrial CO oxidation, the paper should not be considered impactful. We note that our study indeed does not focus on any potential industrial aspects of CO oxidation to CO₂ (and we never stated it was!): on the contrary, we describe a completely new phenomenon of CO oxidation by nitrate in the micropores of a zeolite at unprecedentedly low temperatures: uniquely, completely inorganic abiotic system can perform catalytic chemistry that was previously observed under mild

conditions only in alive organisms which utilize CO oxidation by nitrate to produce energy for Life. Furthermore, search for traces of CO₂ or carbonate minerals on Mars from CO oxidation by either nitrate or perchlorate minerals is ongoing: CO₂ is thought to be produced by live organisms/bacteria – temperatures on Mars reach extremely low temperatures (from -150 to – 30 °C) and our study has clear potential implications for the ability of abiotic system to perform this reaction at low temperatures.

We provide deep spectroscopic characterization of previously unobserved NO_x chemistry in zeolites, and, with the aid of DFT calculations, suggest a pathway for this unusual transformation to occur.

1. Regarding the structure of this manuscript, the authors extensively discussed the formation and complexation processes of NO⁺ species in H-SSZ-13 in the first half of the manuscript, while the second half elaborated the oxidation of CO by NO₂. However, the proposed mechanism of CO oxidation has little to do with NO⁺ (Figure 4). Please the authors illustrate the role of NO⁺ in the NO₂ oxidation of CO.

Response: We had to provide deep insight into N_xO_y chemistry on zeolite under ambient and cryo-conditions in order to understand the complex infra-red patterns of N_xO_y and CO species we see at low temperatures. Without this knowledge, we couldn't have identified the reactions that occur under those conditions. Some of the described chemistry was previously unknown. That is beneficial not only for our study but to multiple communities interested in N_xO_y transformations (atmospheric chemistry, outer space chemistry, environmental chemistry, catalysis communities). NO⁺ seems to be a spectator species in these transformations: the key point is 1). Reaction of NO₂ with the H-zeolite produces NO⁺ and NO₃⁻ through disproportionation of N₂O₄. 2). Reaction of NO with the sub-stoichiometric amounts of oxygen, produces first N₂O₃, which upon reaction with H-Zeolite produces NO⁺ ion without the presence of nitrate. Out of these two cases, the system that contains only NO⁺ ions does not produce CO₂ upon warming-up of (NO⁺-Zeolite + CO) system (for which we also had to unravel unusual spectroscopic interactions with the formation of the first of its kind NO⁺-CO complex), confirming our suggestion that NO⁺ is indeed a spectator species.

However, for the 2nd system, which contains nitrate ions, reaction with CO at low temperature leads to consumption of nitrate and formation of CO₂ at low temperature. Thus, we demonstrate that nitrate is responsible for observed oxidation activity.

2. Figure 1 exhibits the FT-IR spectra of H-SSZ-13 (with Si/Al ~ 12) after NO₂ adsorption. The peaks at ~1870 and 2100 cm⁻¹ should be assigned. Also, I find these peaks become less distinguishable in Si/Al ~ 6 sample (Figure S1). Such difference should be rationalized.

Response: Because we wanted to show the exact changes in the OH region of the parent zeolite upon its interaction with NO₂, in Fig. 1 we show the full zeolite spectrum as the initial spectrum (IR vacuum is used as background and not zeolite itself). In FigS1 we showed the difference spectra upon interaction with NO₂ (zeolite itself used as background). These points are now clarified in the manuscript. The observed bands are typical zeolite bands (see, for example, Fig. 36 in the review by S. Bordiga, C. Lamberti and co-workers “Probing zeolites by vibrational spectroscopies”).

3. In Figure 1, the experimental parameters corresponding to individual line (including red, blue and grey lines) should be detailed. Similar figure legend issue exists almost in every FT-IR spectrum (Figure 2, 3, S1, S4, S5, S6, S7, S8 and S9), which makes some figures (e.g., Figure S6 and S8) almost unreadable.

Response: The gray lines in the FTIR spectra located between the red and blue show the intermediate states of the reaction between those initial and final state. That is very typical presentation for FTIR spectra evolving during adsorption/desorption process monitoring. We have clarified this in figure captions. Furthermore, we have modified Figures S6 and S8 for better readability as suggested by the Reviewer.

4. What is the Si/Al ratio of H-SSZ-13 reported in Figure 3? There are two types of H-SSZ-13 with different Si/Al ratios (i.e., 12 and 6) investigated in this study, and the CO oxidation performance of these two types of H-SSZ-13 should be reported and compared.

Response: The Si/Al ratio of the sample in Fig. 3 has been updated. It is the same as for all other figures in the main text ~ 12 . Si/Al ratio does not change the properties of zeolite structures: it only changes the amount of Bronsted acid protons in the zeolitic micropores. However, the point was to show that similar $NxOy$ adsorption intermediates form in H-SSZ-13 with a different Si/Al ratio which is the case.

5. In the caption of Figure 2, the adsorption temperature was quoted as 100 K, while that mentioned in the main text (Lines 133-136) was 77 K. Such discrepancy should be addressed.

Response: Thanks for pointing it out! We updated the correct temperature for Fig. 2 throughout the text.

6. Lines 136-137: It reads "... adsorption of NO unambiguously and selectively produces the NO^+ -NO complex. Figure 3A)". However, Figure 3 has nothing to do with NO adsorption.

Response: Thanks for catching this typo! Of course, we meant Fig. 2A. The text has been updated.

Reviewer #2 (Remarks to the Author):

This paper reports that NO_2 adsorption on zeolite disproportionate to form a NO^+/NO_3^- complex, and that NO_3^- acted as an oxidant for CO to produce CO_2 at -140 degree C. The reaction mechanism was discussed by in-situ FTIR and DFT calculation. The CO oxidation proceeded in the absence of noble metal nanoparticles or transition metal oxides at very low temperature, although the reaction stopped at high temperature due to desorption of CO from zeolite surface at high temperature. The phenomenon is very interesting but the discussion on FTIR results is still unclear. Therefore, I think that revision is required for the following points.

Response: We are grateful to the Reviewer for taking the time to evaluate our manuscript.

1) In Figure 2, several wavenumbers are shown but assignment was not completed for all the peaks mentioned in the figure. Figure captions should be clearly written to indicate which color corresponds to which state (e.g., NO^+/NO_3^- complex, after CO adsorption, etc.). In Figure 2A, the peak at 2172 cm^{-1} was assigned to NO^+ in NO^+/NO_3^- and was shifted to 2219 cm^{-1} after CO

adsorption. Although the authors did not mention the peak corresponding to NO_3^- (1650 cm^{-1} from Fig. 3) in this figure, the NO_3^- peak seems to not change. Why only NO^+ species was affected by CO adsorption while DFT calculation suggests that NO_3^- is located near the CO adsorbed on the zeolite. The peak at 2140-2150 cm^{-1} should be mentioned.

Response: We thank the Reviewer for this comment. We have included clear description in the Figure 2 caption. N_2O_4 disproportionation produced NO^+ in the cationic positions of zeolite. NO^+ is a cation that can interact with electron-rich CO molecule, producing $\text{NO}^+\text{-CO}$ complex. $\text{NO}_3^-/\text{HNO}_3$ do not interact with CO to any measurable extent and, thus, is not affected by CO presence. 2140-2150-2170 cm^{-1} bands include a complex set of bands for NO^+ , $\text{H}^+\text{-CO}$ (~2174) and $\text{NO}^+\text{-CO}$ complex.

2) In Figure 2B, the authors discussed the NO adsorption onto NO_2 -preadsorbed zeolite and the formation of NO^+/NO complex. I could not follow how the NO^+/NO complex formation relates to the CO oxidation using nitrates adsorbed on zeolites.

Response: We had to provide deeper insight into N_xO_y chemistry on zeolite SSZ-13 (which is another goal of this study of ours) in order to understand the infra-red patterns of NO^+ interactions with typical probe molecules such as CO and NO. Although $\text{NO}^+\text{-NO}$ complex formation is not directly related to CO_2 formation, it does provide a valuable insight into ability of NO^+ to further participate in new bond formation via interaction with CO and NO probe molecules.

In p. 5, line 6 from the bottom, Figure 3A should be Figure 2B?

Response: Yes, thank you!

3) In Figure 3, CO is oxidized to CO_2 , NO^+ peak was shifted to 2172 cm^{-1} again, indicating CO was consumed, and the peak corresponding to CO_2 was newly observed. At the same time, the NO_3^- peak (1650 cm^{-1}) decreased, indicating NO_3^- play a crucial role for CO oxidation. According to proposed mechanism shown in pp. 9-10, HNO_2 is supposed to be formed after the CO oxidation. Did they observe the formation of HNO_2 by FTIR? Again, the peak at ca. 1750 cm^{-1} was increased by not explained the reason.

Response: We thank the Reviewer for bringing it up. We looked carefully at the literature on HNO₂ bands – although surprisingly no papers could be found studying HNO₂ adsorption on any solid materials (presumably due to difficulties associated with producing it cleanly), studies for gas/liquid-phase HNO₂ bands could be found – cis and trans isomers exist with N-O stretch of the trans-isomer around somewhere ~ 1,700 cm⁻¹. Furthermore, the stretch could be identified around ~1,260 characteristic of HON bend HNO₂ species (see the inset and comparison with the published data) (we have added this to the text and additional SI S12 graph showing this region).

Figure S12. *In-situ* FTIR of HON bending region during increase of temperature from 100 K (red line, described in Figure 2A, H-SSZ-13 with Si/Al ~12) to 135 K. At this temperature CO₂ starts to evolve and the temperature is held until CO₂ reaches the maximum level (~2 minutes). The inset shows the magnified 2,400-2,000 cm⁻¹ region. Zeolite pellet itself was used as IR background.

Figure 2. Gas-phase infrared spectra (a) after 120 min during the reaction of adsorbed HNO_3 (initially 9×10^{18} molecules) with 6.5×10^{16} molecules cm^{-3} NO, (b) after subtraction of the ν_{11} band of N_2O_4 from (a), showing the ν_3 band of gas phase *trans*-HONO whose intensity corresponds to a concentration of $\sim 2 \times 10^{14}$ molecules cm^{-3} , and (c) of NO alone; the small peak at 1263 cm^{-1} is due to trace impurity levels of HONO in the NO.

Representative Figure for the Reviewer, showing that HON bending vibration of HNO_2 is observed at $1,260 \text{ cm}^{-1}$. (J. Phys. Chem. A, Vol. 104, No. 43, 2000).

4) In p. 8, the authors mentioned that CO is adsorbed at the Bronsted acid sites. Is there any possibility that CO is adsorbed at the Lewis acid sites on the zeolite?

Response: We tried to model a complex of CO with a framework Al center, but the Al-C distance in the optimized structure was above 330 pm, implying very weak interaction.

5) In Zeo1/CO/HNO3 in Mechanism C, H is missing.

Response: Thank you for pointing it out. The missing H atom was added.

Reviewer #3 (Remarks to the Author):

This manuscript reports the discovery of low-T CO oxidation in NO₂-activated SSZ-13 zeolites using FTIR and DFT studies, along with proposed mechanisms and energetics of CO oxidation. The manuscript (with the exception of few grammatical errors and incomplete sentences) clearly lays out the details of the formation of the active NO⁺ species on the zeolite and CO adsorption characteristics. The work is new and interesting.

Response: We thank the reviewer for their effort in reviewing this study.

My primary concern lies with the DFT studies since they play a large role in interpretation of experimental results. While it is encouraging that DFT results agree with experiment in general, the authors must note that PBE+dispersion may not be the best choice of theory for SSZ-13 (See: <https://doi.org/10.1002/cctc.201900791>). This is especially important for binding energies, where it is necessary to verify whether CO indeed binds weakly to the zeolite-bound NO⁺. This will also help further verify the proposed mechanism(s) by examining whether the energetics of the four pathways are sensitive to the choice of level of theory.

However, DFT errors vary with the properties as well (vibrational frequencies, binding energies), making it difficult to choose the most physically appropriate level of theory. I therefore recommend the authors perform at least a small subset of all calculations using a functional more suited for these systems (e.g. hybrid functionals, SCAN) to verify that their interpretation of experimental observations are still valid.

Response: Following the suggestion, we performed several test calculations employing one hybrid (HSE06) functional and one meta-GGA (SCAN) functional. The results with the three functionals (PBE+D2, SCAN, and HSE06) are summarized in Tables S2 and S3 and in a new paragraph entitled “Verification of the employed PBE+D2 functional”, which is included in the Supplementary Materials.

The calculated binding energies of CO and HNO₃ in the ZEO1/CO/HNO₃, as well as for NO and NO₂ in the complexes Zeo/NO⁺-NO and Zeo/NO⁺-NO₂, respectively, showed that the results obtained with PBE+D2 are close to the values obtained with HSE06 and/or SCAN functionals. We also compared computational results for reaction energy of the CO oxidation by HNO₃ in the gas phase to CO₂ and HNO₂ with experimental value derived from the standard experimental enthalpies. All three functionals gives values close to the experimental estimate, as for the PBE+D2 functional the difference from the experiment is only 1 kJ/mol.

The calculated vibrational frequencies obtained with PBE+D2 functional were notably closest to the experimental values, while HSE06 and SCAN gave significantly higher (by 100-200 cm⁻¹ for the HSE06 and 200-300 cm⁻¹ for the SCAN) vibrational frequencies. This comparison also suggests that the computational approach, used in the present work is relevant for the studied process.

In order to check whether the proposed most plausible reaction pathway for CO oxidation by HNO₃ in the zeolite is functional dependent we calculated all the barriers at HSE06 level. Our results showed that all the HSE06 reaction energies are very close to PBE+D2 values, however the reaction barriers are systematically shifted to higher values by 50-80 kJ/mol in comparison to values calculated with PBE+D2 functional. However, at both levels of theory the order of barrier heights was the same: TS₂>TS₃>TS₁. Relevant citations have also been added.

Minor concerns:

1. In Figure 1, how do the authors interpret the new OH stretch at 3667cm⁻¹?

Response: Most likely, it belongs to -OH stretch of HNO₃ molecules in zeolite micropore. It is now included in the text.

2. Line 53 – Sentence is incomplete

Response: We have fixed this mistake.

3. Scheme 1 is not charge-balanced (H⁺-zeolite on LHS?)

Response: Thank you for catching it! We added the missing H atom.

REVIEWERS' COMMENTS

Reviewer #1 (Remarks to the Author):

The authors have carefully addressed the raised issues in the revised manuscript and I would like to suggest the acceptance of this paper.

Reviewer #2 (Remarks to the Author):

The manuscript has been revised according to my comments and can be accepted.

Response to reviewers' comments

Manuscript Number: NCOMMS-20-30796.

Title: Biomimetic CO oxidation below -100 °C by a nitrate-containing metal-free microporous system

Author(s): Konstantin Khivantsev, Nicholas R. Jaegers, Hristiyan A. Aleksandrov, Libor Kovarik, Mirosław A. Derewinski, Yong Wang, Georgi N. Vayssilov and Janos Szanyi

REVIEWERS' COMMENTS

Reviewer #1 (Remarks to the Author):

The authors have carefully addressed the raised issues in the revised manuscript and I would like to suggest the acceptance of this paper.

Reviewer #2 (Remarks to the Author):

The manuscript has been revised according to my comments and can be accepted.

Our response: We thank the Reviewers for taking the time to evaluate our study.